# Removal of Environmentally Harmful and Hardly Degradable Pharmaceuticals Sulfamethoxazole, Diclofenac, and Cetirizine by Adsorption on Activated Charcoal



**Daniela Pavúková [1], Lucia Fašková [1], Eva Melníková [1], Emília Mališová [1], Ján Híveš [1], Ladislav Štibrányi [1], Pavol Hudec [1], Monika Naumowicz [2],\* and Miroslav Gál [1],\***

[1]   Faculty of Chemical and Food Technology, Slovak University of Technology in Bratislava, Radlinského 9, 812 37 Bratislava, Slovakia
[2]   Department of Physical Chemistry, Faculty of Chemistry, University of Bialystok, Ciolkowskiego 1K, 15-245 Białystok, Poland
\*   Correspondence: monikan@uwb.edu.pl (M.N.); miroslav.gal@stuba.sk (M.G.)

**Abstract:** The removal of three environmentally harmful and hardly degradable pharmaceuticals, namely sulfamethoxazole, diclofenac, and cetirizine, from aqueous solution by the adsorption onto two types of activated charcoals (WSCl2 and HWOH) was investigated. The volume of micropores and mesopores in two charcoals was the main property affecting removal efficiencies. Using microporous WSCl2 as an adsorbent, higher removal efficiencies were achieved for all chosen pharmaceuticals. The highest removal efficiency was recorded in the case of sulfamethoxazole (79%). A direct correlation between $\log K_{ow}$ and removal efficiencies and between the solubility of pharmaceuticals and removal efficiencies was not found. The adsorption behavior of individual pharmaceutical solutions can be described by the pseudo-second order kinetic model. The parameters obtained from the kinetic model show that the adsorption rate on HWOH was higher than on WSCl2. However, the amounts of adsorbed pharmaceuticals were lower on HWOH than on WSCl2, which can be linked to the textural difference between the charcoals. In the mixture consisting of all three compounds, overall removal efficiencies were lower than in the case when individual pharmaceuticals were present in the solution. Results also indicate that a certain fraction of the micropores can only be occupied by the smallest compound in the mixture (sulfamethoxazole).

**Keywords:** activated charcoal; adsorption; pharmaceuticals; WWTP; $\log K_{ow}$

## 1. Introduction

Pharmaceutically active compounds (PhACs), their residues, and metabolites occur in the surface, drinking and groundwater all over the world [1]. The reason behind their presence is the inability of the methods used in wastewater treatment plants to efficiently remove certain types of PhACs. Removal efficiencies vary from 0% to 100% depending on the compound [2]. Therefore, the development of a new, more efficient technology for the elimination of such compounds is a must. There is a large quantity of technologies that have been tested in the last decades. The most significant are advanced oxidation processes (AOPs, e.g., sonochemistry, electrooxidation, photochemistry, ozonation) and sorptive removal [3–5]. A major drawback of the former is the uncertainty of toxic or biologically active transformation product formation [6]. On the contrary, the latter does not generate pharmaceutically active or toxic products [7]. Adsorption on activated charcoal (AC) is a powerful and flexible method for water purification, and it has been widely used to prevent environmental pollution. AC is a form of carbon that is microcrystalline and has a high degree of porosity and large surface area, which allows extensive adsorption of various substances from liquids [8–11].

In our work, we have studied the adsorption of diclofenac, sulfamethoxazole, and cetirizine. These are among the most detected PhACs in the environment [12]. Diclofenac is a widely used nonsteroidal anti-inflammatory drug found in the water environment [1]. It reduces inflammation and relieves pain and can be administrated orally or applied to the skin. It is detected in surface water [13], groundwater [14] but also in drinking water [15]. At the concentration of only 5 ug/L, renal lesions and alterations of the gills were observed in rainbow trout [16]. At the concentration of 0.17 mg/L, changes in the feeding behavior of medaka fish occurred [17]. Sulfamethoxazole is a widely used sulfonamide antibiotic applied to treat bacterial infections of the urinary tract or bronchitis. It is also extensively used in animal husbandry [18]. Due to its high water solubility, sulfamethoxazole is often found in almost all water bodies all over the world [1,13,15]. Generally, antibiotics in the water environment affect algae more than fish or crustaceans, but the main concern arises from the development of antibiotic resistance in bacteria [19]. Cetirizine is a second-generation antihistamine used to treat allergic reactions. Therefore, high concentrations are found in water bodies, mainly in spring [20]. It has been proved that long-term exposure to cetirizine induced biochemical alterations in the metabolic activity and oxidative stress-related markers in mussels [21].

The process of the adsorption of PhACs on activated charcoal depends on several factors, including not only the properties of AC but also the physical and chemical properties of the pharmaceuticals. The PhACs properties that have the strongest impact on adsorption efficiency are hydrophobicity/hydrophilicity (log $K_{ow}$), molecular size, pKa, pH, and solubility [22]. Pharmaceuticals such as ibuprofen [23] and diclofenac, which have higher log $K_{ow}$, are adsorbed better than substances with low log $K_{ow}$, such as sulfamethoxazole [24]. Moreover, the less PhAC is soluble in water, the easier it is removed. Carbamazepine, sulfamethoxazole and ketoprofen, which have low solubility in water, were removed faster than highly soluble terbutaline, fluoxetine and metoprolol [25].

The most important properties of AC are BET surface area, pore size, distribution and their surface functional groups [26]. The adsorption rate on porous materials such as activated charcoal is highly related to their porous features. We recognize three different pore types depending on their volume. Micropores have diameters less than 2 nm, and they are mainly responsible for the adsorption of molecules. Mesopores have diameters between 2 nm and 50 nm and act as a transportation network. Lastly, macropores with a diameter above 50 nm serve for the entrance of the molecule into AC [26]. Well-developed mesopore structure is said to be crucial for providing access for the adsorbate into inner porosity. Therefore, higher adsorption rates are expected in mesoporous materials. Nonetheless, micropores are intended to be active sites of adsorption; but they must be spacious enough to accommodate the molecule of the adsorbate [27]. Research shows that AC with a microporous structure has higher removal efficiency (80% to 100%) in removing smaller molecules such as ibuprofen, paracetamol acetylsalicylic acid and clofibric acid. In contrast, the mesoporous structure was found to be essential to retain bulkier compounds such as iopamidol [8]. Galhetas et al. [28] also found a connection between the porous structure of AC and the adsorption rate. Their results pointed out that for the adsorption of acetaminophen, the mesoporous structure did not have a positive effect on the contrary presence of a developed micropore structure was the key factor that controlled the rate of adsorption in the system. Ji et al. [29] corroborate the effect of different pore structures on the disparity of adsorption patterns in their research. Adsorption of bulky tetracycline and tylosin was much lower for the microporous AC in contrast to low-sized sulfamethoxazole.

It is important to point out that medicinal products usually do not occur in the environment as isolated substances. A variety of different compounds is used simultaneously in human and veterinary medicine. Moreover, most medicinal products are partially transformed into metabolites creating a multi-component mixture of parent compounds and metabolites [30]. Adsorption of multiple pharmaceuticals mixture is affected by mutual interaction and competition between these substances. Thus, the amount of adsorbed PhACs can differ [31]. When adsorbed from the mixture (carbamazepine, sulfamethoxa-

zole, and trimethoprim), the adsorption of sulfamethoxazole was most negatively affected by the presence of other components since having the lowest log $K_{ow}$ [32]. In addition, when PhACs are in a mixture, AC creates a sieving effect meaning smaller molecules such as caffeine or carbamazepine were adsorbed more than molecules with higher volumes than these PhACs (for example, diclofenac) [33–35]. Overall, there are many benefits to utilizing activated carbon to remove pharmaceuticals from waste streams. It is an easy, highly effective, and ecologically friendly approach that may be used in batch as well as continuous processes. However, higher manufacturing costs and a frequent requirement for regeneration could limit its overall effectiveness.

In this work, we focus on how selected properties of chosen pharmaceuticals (solubility, molecular size, log $K_{ow}$) and activated charcoals (surface area, pore distribution, textural properties) affect the overall removal efficiencies. Two types of AC were investigated in the batch system using either a single pharmaceutical experiment or a mixture of all pharmaceuticals to compare the overall AC performance. Pharmaceuticals were chosen according to their occurrence in the environment, and their concentration was set to be like the environmentally relevant concentrations while still above the threshold of our analytical system. There are some discrepancies in the literature among obtained results, as stated before, and therefore, elucidation of the association of the above-mentioned parameters is important for future environmental applications.

## 2. Materials and Methods

### 2.1. Materials

Sulfamethoxazole (SMX), diclofenac (DCF), and cetirizine (CET) were purchased from Sigma-Aldrich and were of analytical purity; HPLC grade acetonitrile was purchased from VWR International.

### 2.2. Single Pharmaceutical Batch Experiments

Single PhAC batch experiments were conducted under magnetic-controlled stirring (500 rpm). Solution of individual pharmaceuticals in distilled water with a concentration of 2 mg/L was placed on a magnetic stirrer, and 10 mg/L of activated charcoal was added and stirred. Samples for HPLC analysis were taken at (1, 2, 3, 5, 10, 20, 30, 40, 50 and 60) minutes. The pH during the experiment was not adjusted and remained almost unchanged during the whole experiment.

### 2.3. Kinetic Experiments

Kinetic studies were basically identical to the single pharmaceutical batch experiments except for the timescale. Solutions were mixed until the system reached equilibrium. All samples were analyzed by HPLC every 5, 20, 40, 60, 80, 120, 180 and 240 min.

### 2.4. Mixture Batch Experiments

Batch adsorption experiments of the mixture of three PhACs, diclofenac, sulfamethoxazole, and cetirizine, were also investigated. To the mixture with a concentration of 2 mg/L of each PhAC, 30 mg/L and 10 mg/L of activated charcoal was added. Samples for HPLC analysis were taken at (1, 2, 3, 5, 10, 20, 30, 40, 50 and 60) minutes. The mixture was stirred at 500 rpm.

### 2.5. Analytical Detection and Quantification

The analysis of the samples was conducted using high-performance liquid chromatography using a chromatograph Thermo Fisher Scientific UltiMate 3000 with a DAD detector. The used column was Reprosil 100 C18, 5 μm, 250 mm × 4 mm. The mobile phase for isocratic mode was 50:50 ($v/v$) acetonitrile/water and phosphate buffer at 1 mL/min.

The removal efficiency was calculated according to the following equation (Equation (1)):

$$removal\ efficiency\ (\%) = \frac{c_0 - c}{c_0}\ 100\% \tag{1}$$

where $c_0$ and $c$ (mg/L) are the initial concentration and concentration at the t $\neq$ 0 of PhAC, respectively.

*2.6. Characterization of Charcoals*

Two charcoals with different properties were chosen. HWOH, manufactured by Hrušovské chemické závody, was made of wooden sawdust and activated by $ZnCl_2$. WSCl2, manufactured by SLZ Hnúšťa, was made of hard beech wood and activated by KOH. The temperature of pyrolysis was between 500 °C to 900 °C in both cases. The texture properties of active carbon samples were studied by physical adsorption of nitrogen at −196 °C using ASAP-2400 (Micromertics). Before the analysis, all three samples were evacuated overnight at the temperature of 150 °C under the vacuum of 2 Pa. Specific surface area (SBET) was obtained using conventional BET isotherm ($p/p_0$ = 0.05–0.3). The external surface area with the surface of mesopores $S_{t(mesopores)}$ and the volume of micropores ($V_{micropores}$) were calculated from the t-plot using the Harkins—Jura master isotherm. Total pore volume ($V_{total}$) was determined from the volume of nitrogen adsorbed at a relative pressure of $p/p_0$ = 0.99. The $pH_{pzc}$ of both ACs, where $pH_{pzc}$ is the pH value at which the net surface charge of the adsorbent equals zero, was studied by the pH drift method according to work by Jedynak & Charmas [35] using a Methrom 913 pH meter with combined glass electrode with SD = 2%.

## 3. Results

Firstly, we have characterized the textual properties of both charcoals. Two kinds of activated charcoal were used for the batch experiments depicted as WSCl2 and HWOH. Both ACs were in powdered form. FTIR spectra (not shown) show in both cases standard and similar stretching vibration in wavelength ranges of 1040–1240, 1560–1570, 1715, 2050 and 3495–3630 cm$^{-1}$. Based on the FTIR assignments of functional groups on carbon surfaces, these bands can be assigned to stretching of C-OH (phenolic), quinones, carboxylic salts, -OH groups and alcohols, respectively. In the case of HWOH, these bans are more intense compared to the WSCl2 sample indicating that more specific groups are present at the respective charcoal surface. The X-ray diffraction pattern showed that both samples had an amorphous structure with a small portion of crystalline graphite (trigonal) in the case of HWOH. In the case of WSCl2, the peak that corresponds to the so-called supercubine carbon structure is present. This confirms BET observation that in the case of WSCl2 charcoal more compact (microporous) structure is present.

Textural properties of ACs obtained from BET analysis are listed in Table 1.

**Table 1.** Main textural properties of AC samples.

| AC | $S_{BET}$ [a] (m$^2$·g$^{-1}$) | $V_{micropores}$ [b] (cm$^3$·g$^{-1}$) | $S_{(t)mesopores}$ [c] (m$^2$·g$^{-1}$) | $V_{total}$ [d] (cm$^3$·g$^{-1}$) |
|---|---|---|---|---|
| WSCl2 | 989 | 0.364 | 293 | 0.648 |
| HWOH | 1.074 | 0.047 | 977 | 0.980 |

Note(s): [a] Specific surface area (BET method); [b] Volume of the micropores (t-plot method); [c] Mesopores surface area (t-plot method); [d] Total pore volume evaluated at $p/p_0$~0.99.

As can be seen from Table 1, ACs used for the experiments have different textural properties. HWOH has a higher BET surface area than WSCl2, but its volume of micropores is noticeably smaller. BET results are also confirmed by SEM images of the samples (Chart 1). HWOH charcoal is characterized by larger pores compared to WSCl2.

As mentioned above, the micropore area contributes the most to the adsorption capacity, and mesopores serve more as a transport network. However, the micropores must be large enough to contain the molecule of an adsorbate. If adsorbate is bulkier, AC with a more mesoporous structure is suitable for its removal [32].

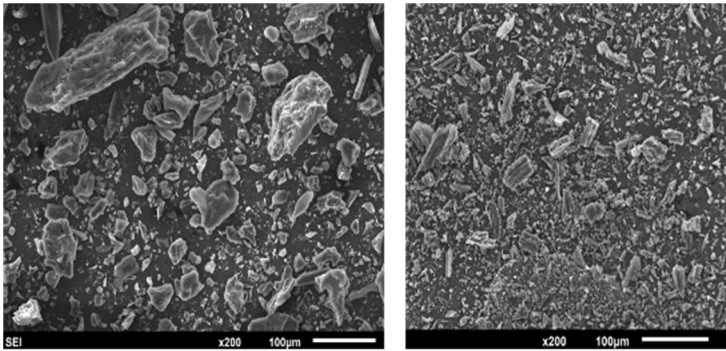

**Chart 1.** SEM images of HWOH (**left**) and WSCl2 (**right**).

Although the pH was not adjusted during experiments to avoid the unwanted influence of buffer on the adsorption process, the pH values of PhACs solutions and their suspensions with AC were controlled during the experiment, and the obtained pH values are shown in Tables 2 and 3 for WSCl2 and HWOH, respectively.

**Table 2.** pH values of PhACs solutions before and after adsorption using WSCl2. SD = ± 2%.

| Sample | pH of PhAC Solution | pH of AC Suspension before Adsorption | pH of AC Suspension after Adsorption |
|---|---|---|---|
| diclofenac | 6.45 | 6.08 | 6.17 |
| sulfamethoxazole | 6.30 | 5.71 | 6.70 |
| cetirizine | 5.64 | 5.77 | 6.13 |
| mixture | 5.59 | 5.45 | 5.72 |

**Table 3.** pH values of PhACs solutions before and after adsorption using HWOH. SD = ± 2%.

| Sample | pH of PhAC Solution | pH of AC Suspension before Adsorption | pH of AC Suspension after Adsorption |
|---|---|---|---|
| diclofenac | 6.45 | 6.14 | 6.02 |
| sulfamethoxazole | 6.30 | 5.86 | 5.78 |
| cetirizine | 5.64 | 5.55 | 5.36 |
| mixture | 5.59 | 5.55 | 5.38 |

As can be seen from both Tables 2 and 3, the pH is quite stable during the adsorption of all samples on both ACs used, and the influence of these changes can be neglected. This finding is important because considering the pKa values of pharmaceuticals used for adsorption experiments ($pK_{a(DCF)} = 4.15$; $pK_{a1(SMX)} = 1.6$, $pK_{a2(SMX)} = 5.7$; $pK_{a1(CET)} = 2.7$, $pK_{a(CET)2} = 3.57$, $pK_{a3(CET)} = 7.56$) DCF and SMX were negatively charged in all experiments and CET was in the form of zwitterion in all experiments. Another factor that is important for the adsorption of micropollutants on ACs are $pH_{pzc}$ values of WSCl2 and HWOH. There were determined to be **2.8** and **3.2**, respectively, i.e., during all experiments, both ACs were negatively charged. This means that the electrostatic forces act preferably against the adsorption forces of sulfamethoxazole and diclofenac on the studied ACs. However, according to the pH values of individual solutions, $pK_a$ of PhACs and $pH_{pzc}$ of ACs, the differences between electrostatic repulsive forces between individual PhACs and ACs are not very high.

### 3.1. Single Pharmaceuticals Batch Experiments

Single batch experiments were carried out to determine the adsorption of individual pharmaceuticals onto two ACs to evaluate their removal efficiency. We focused on the removal capabilities of chosen ACs to further compare single PhAC and mixture systems.

In the beginning, the time dependence of the adsorption of individual PhACs on each AC was studied. The aim of these experiments was to estimate the minimal time that is necessary for respective PhACs to be adsorbed on the individual ACs. In Figure 1, the dependences of removal efficiency of diclofenac, sulfamethoxazole, and cetirizine on reaction time using WSCl2 (Figure 1a) and HWOH (Figure 1b) as adsorbent are plotted.

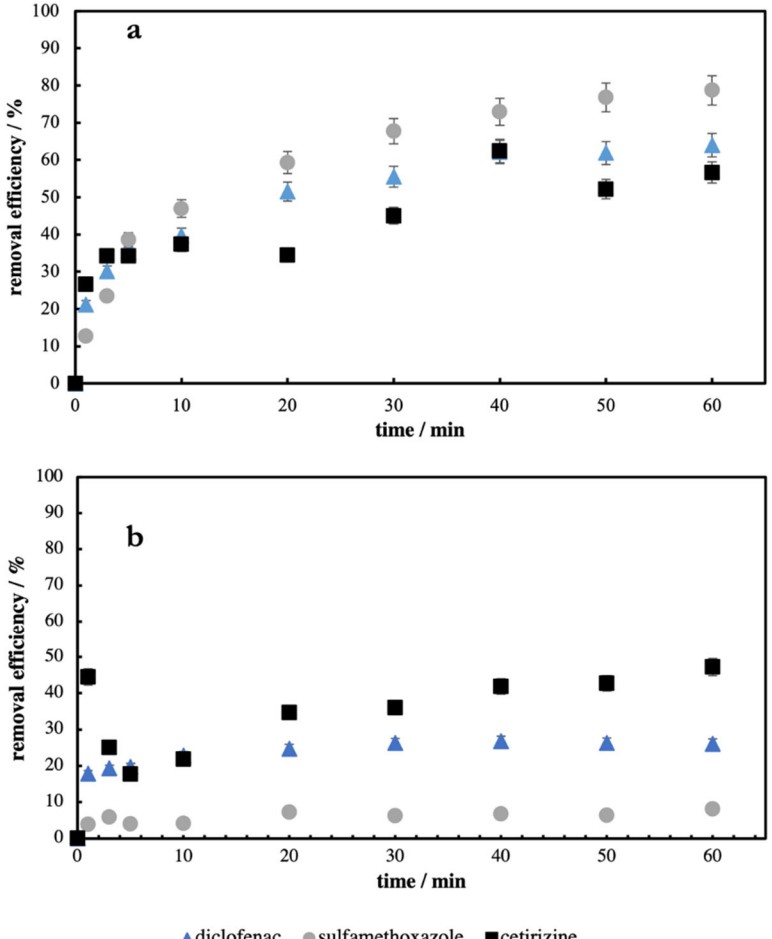

**Figure 1.** Dependence of removal efficiency of diclofenac, sulfamethoxazole and cetirizine on the reaction time using WSCl2 (**a**) and HWOH (**b**) as adsorbent.

As can be seen from Figure 1, after 50–60 min, the plateau (broad maximum) at all curves is reached. It means that the adsorption of individual PhACs on selected ACs is finished after ca. 50–60 min, and longer experiments would not increase the amount of removed (adsorbed) PhACs. This fact was confirmed during/by kinetics studies. The time of our further experiments was, therefore, set to 60 min because the removal efficiency reached its maximum and did not increase after that time.

In our experiments, the only bulkier molecule was cetirizine (see Table 4).

**Table 4.** Percentage of removal of diclofenac, sulfamethoxazole, and cetirizine from a single PhAC solution in 60 min and their chosen physicochemical properties.

| PhAC | log $K_{ow}$ [a] | $s_w$ [b] (mg.dm$^{-3}$) | AC | Removal Efficiency % |
|---|---|---|---|---|
| diclofenac | 4.51 | 2.37 | WSCl2 | $62 \pm 3$ |
| | | | HWOH | $21 \pm 2$ |
| sulfamethoxazole | 0.89 | 610 | WSCl2 | $79 \pm 5$ |
| | | | HWOH | $8 \pm 2$ |
| cetirizine | 2.98 | $6.96 \times 10^4$ | WSCl2 | $57 \pm 3$ |
| | | | HWOH | $47 \pm 6$ |

Note(s): [a] Octanol–water partition coefficient (Pubchem database, 2022); [b] Solubility in water at 25 °C (Pubchem database, 2022).

This PhAC was the only one in which similar removal efficiency in the designated time in the case of both ACs (57% and 47% for WSCl2 and HWOH, respectively) was observed. In the case of other PhACs, differences in AC performances were more visible. Higher removal efficiencies were achieved using WSCl2 as an adsorbent for all chosen pharmaceuticals (see Figure 1a,b). The highest removal efficiency was achieved when sulfamethoxazole was used in the system (79%). In the case of all PhACs, the adsorption rate was fastest during the first 10 min of the time experiment compared to the rest of this experiment. After 10 min, the adsorption rates in all cases decreased (see Figure 1). Because adsorption rates are closely connected to the surface characteristics of the adsorbent, these results can be caused by such different textural properties. During the adsorption of cetirizine on HWOH, we observed a steep increase in removal efficiency in the first minutes of the experiment (44%). Then the adsorbed amount of PhAC decreased in the third and fifth minutes and began to increase again until the end of the experiment (50–60 min). After this time, the amount of cetirizine on HWOH reached a similar value (47%) as after the first minute. This behavior was reproducible ($\pm$ 5%) several times in this adsorption system. It can be explained as the quick uptake of cetirizine in mesopores and its subsequent relatively fast desorption followed by next re-adsorption in micropores during the rest of the experiment caused by HWOH's small volume of micropores for cetirizine to be moved inside them.

Since the net surface charge of both ACs was negative, negative repulsions between ACs and PhACs are expected. However, from Figure 1a,b, one can see that this effect is not the most significant. The removal efficiency of, e.g., CET was highest in the case of HWOH and the lowest in the case of WSCl2. The adsorption of SFX was, in opposite to CET, highest on WSCl2 and the lowest on HWOH and the removal efficiency of DIC was in the middle between CET and SFX on both ACs. However, all ACs and PhACs were under the same conditions during experiments which means that other, stronger effects must be considered to explain such behavior. It seems that the zwitterion form of CET also plays a role in its adsorption on ACs.

Some authors [24,36] found a direct correlation between log $K_{ow}$ and removal efficiency. The larger the log $K_{ow}$ value (higher hydrophobicity), the higher the removal efficiency. According to our results, when WSCl2 was used, log $K_{ow}$ of chosen PhACs has not played a significant role in affecting removal efficiencies or removal rates, respectively. For example, 62% of diclofenac (log $K_{ow}$ = 4.51) and 79% of sulfamethoxazole (log $K_{ow}$ = 0.89) was removed under the same conditions. The same can be said for the impact of solubility. Contrary to the study of Nebout et al. [25], where molecules less soluble in water were adsorbed more easily, we found no clear evidence of such behavior. Removing efficiency of cetirizine, the most soluble compound from chosen PhACs, achieved the lowest values (47%) from all PhACs using WSCl2. However, when HWOH with sulfamethoxazole with rather poor water solubility was used, the worst adsorption was observed (8%). In this case, the low log $K_{ow}$ value of sulfamethoxazole could play a more significant role than the overall solubility of this pharmaceutical.

The obtained data were studied using several kinetics models. The best fit was found for the pseudo-second order model (Equation (2)):

$$\frac{t}{q} = \frac{1}{k_2 q_e^2} + \frac{t}{q_e} \tag{2}$$

where $t$ is time (min), $q$ is adsorption capacity (mg/g), $k_2$ is pseudo-second order rate constant (g/mg/min), and $q_e$ is adsorption capacity at the equilibrium (mg/g). Data from the fitting procedure for all PhACs and both ACs used in our experiments are summarized in Table 5 and plotted in Figure 2. Table 5 shows second-order rate constants $k_2$, calculated maximum equilibrium uptake $q_{e,cal}$, and multiple correlation coefficient squared $R^2$.

**Table 5.** Kinetic data of adsorption of diclofenac, sulfamethoxazole, and cetirizine on two activated charcoals fitted to the pseudo-second order.

| PhAC | Activated Charcoal | $k_2 \times 10^4$ (g·mg$^{-1}$·min$^{-1}$) | $R^2$ | $q_{e,cal}$ (mg·g$^{-1}$) |
|---|---|---|---|---|
| diclofenac | WSCl2 | 12.6 | 0.9932 | 133.1 |
| | HWOH | 123.1 | 0.9983 | 44.2 |
| sulfamethoxazole | WSCl2 | 7.7 | 0.9966 | 174.9 |
| | HWOH | 318.3 | 0.9913 | 15.8 |
| cetirizine | WSCl2 | 17.1 | 0.9519 | 118.3 |
| | HWOH | 30.7 | 0.9795 | 82.7 |

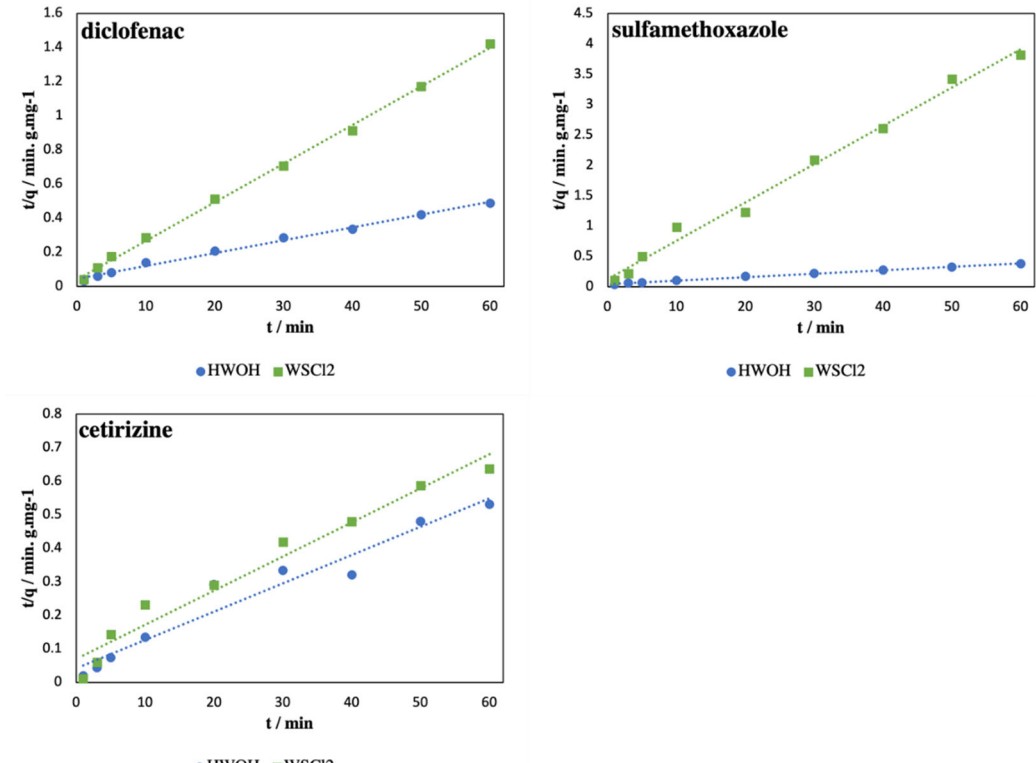

**Figure 2.** Adsorption kinetic plotted as a ratio of time (t) to uptake at given time (q) to time for two adsorbents (WSCl2 and HWOH) and three chosen pharmaceuticals (diclofenac, sulfamethoxazole and cetirizine).

The pseudo-second order kinetics is typically connected with cases where the rate of the direct adsorption process controls the total sorption kinetics. The adsorption rate

constants in the case of HWOH were always higher than in the case of WSCl2, regardless of the pharmaceutical. It means that the adsorption rate on HWOH was higher than on WSCl2. This clearly reflects the presence of mesopores that cause quick transport of adsorbate into the micropores, which is in agreement with the literature data [23,33]. However, the low micropores volume in HWOH caused a lower adsorbed quantity of PhACs (Table 5). Only in the case of cetirizine the rate constants were of the same order for both ACs, and there was the smallest difference in $q_e$ values compared to other PhACs. This indicates that cetirizine was partly accommodated also in the mesopores of HWOH.

### 3.2. Mixture Batch Experiments

After single pharmaceutical measurements, an adsorption experiment with the mixture of three pharmaceuticals, diclofenac, sulfamethoxazole, and cetirizine, was also done, and the removal efficiencies were calculated. Removal efficiencies of diclofenac, sulfamethoxazole, and cetirizine from the mixture of these pharmaceuticals after 60 min on both ACs are summarized in Table 6.

**Table 6.** Removal efficiencies of diclofenac, sulfamethoxazole, and cetirizine from the mixture of these pharmaceuticals after 60 min experiments on 10 mg/L and 30 mg/L of WSCl2 and HWOH, respectively.

| PhAC | log $K_{ow}$ [a] | $s_w$ [b] (mg.dm$^{-3}$) | AC | Removal Efficiency (10 mg/L of AC) % | Removal Efficiency (30 mg/L of AC) % |
|---|---|---|---|---|---|
| diclofenac | 4.51 | 2.37 | WSCl2 | 30 ± 3 | 57 ± 5 |
| | | | HWOH | 8 ± 2 | 21 ± 2 |
| sulfamethoxazole | 0.89 | 610 | WSCl2 | 32 ± 3 | 61 ± 6 |
| | | | HWOH | 4 ± 2 | 9 ± 1 |
| cetirizine | 2.98 | $6.96 \times 10^4$ | WSCl2 | 23 ± 2 | 44 ± 3 |
| | | | HWOH | 19 ± 5 | 33 ± 6 |

Note(s): [a] Octanol–water partition coefficient (Pubchem database, 2022); [b] Solubility in water at 25 °C (Pubchem database, 2022).

As expected, the overall removal efficiencies were significantly lower in the case of a mixture of PhACs than in the case when individual pharmaceuticals were present in the solution using the same amount of adsorbent (Table 6).

Similarly, to the single PhAC experiments, at the beginning of mixture experiments, the time-depended adsorption studies were carried out. In the next Figure 3, the dependences of removal efficiency of diclofenac, sulfamethoxazole and cetirizine mixture on time using WSCl2 and HWOH as adsorbent are plotted.

The overall adsorption of PhACs from their mixture is significantly lower than in the case of a single PhAC system if the same amount of individual PhAC is used (compare Figure 3a,b to Figure 1). Hence we decided to triple the amount of both ACs used in the previous experiments to ensure the same ratio of AC to PhAC (Figure 3c,d). Then the total amount of adsorbed all three PhACs from the mixture was comparable to the amounts adsorbed from the solution where only individual pharmaceuticals were present. It means that usable micropores located at the surface of the respectively activated charcoal are fully occupied by PhACs after ca. 50–60 min of the adsorption process, independent from the number of PhACs present in the system. In the case of WSCl2 charcoal, the competitive behavior between sulfamethoxazole and diclofenac was observed. The amount of diclofenac at the beginning of the adsorption process was higher than the amount of sulfamethoxazole. However, after ca. 40 min, the amount of diclofenac relatively decreased compared to the amount of adsorbed sulfamethoxazole. This behavior was not observed if only one pharmaceutical was present in the system (see Figure 1). This indicates that a certain fraction of the micropores can only be occupied by the smallest compound in the system (sulfamethoxazole). Another result of the mixture experiment is

that, albeit sulfamethoxazole was the most adsorbed, diclofenac was the least affected by the competition of other PhACs. The main reason for such behavior can be that diclofenac has a five times higher log $K_{ow}$ value compared to sulfamethoxazole (Table 6).

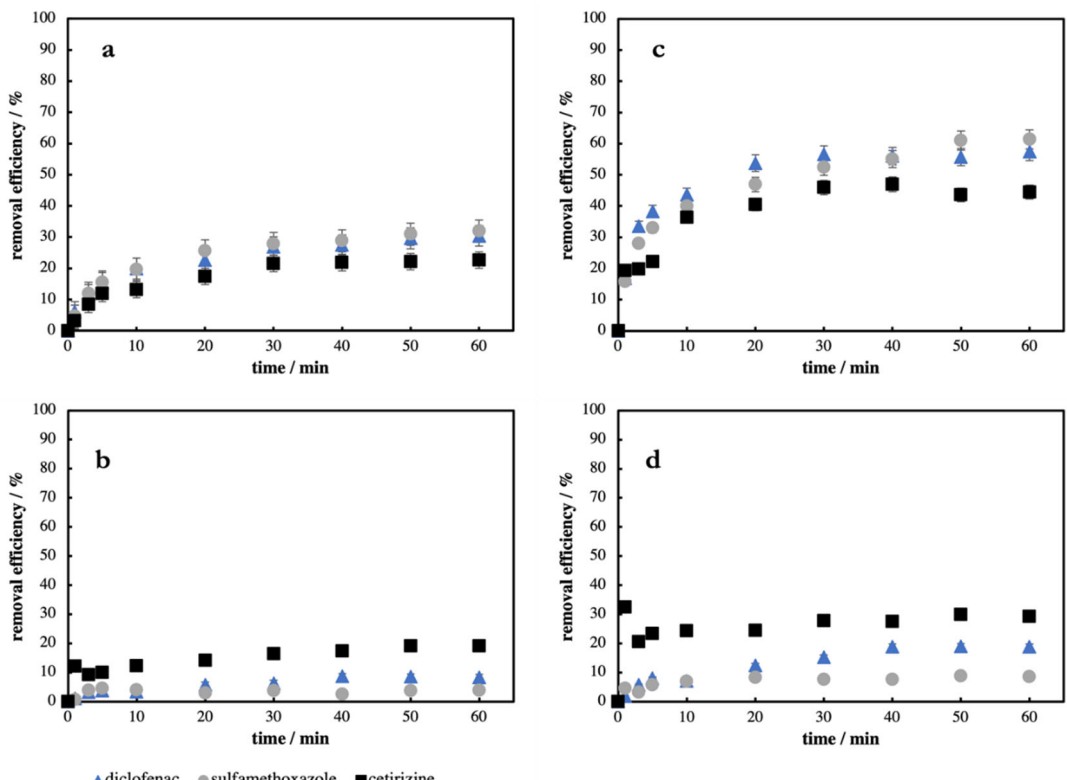

**Figure 3.** Dependence of removal efficiency of diclofenac, sulfamethoxazole and cetirizine mixture from time using 10 mg/L of (**a**) WSCl2 and (**b**) HWOH; (**c**) 30 mg/L of WSCl2 and (**d**) HWOH as adsorbent.

It can be seen from Figure 3 and Table 6 that overall removal efficiencies achieved by using WSCl2 were higher than those using HWOH charcoal. As in a single pharmaceutical solution, cetirizine was adsorbed in larger quantities when using HWOH than WSCl2. Removal efficiencies of sulfamethoxazole and diclofenac using HWOH as an adsorbent were noticeably lower, thus proving the role of mesopores. Basically, we can observe similar trends as in a single PhACs experiment except the one discussed above.

## 4. Conclusions

The adsorption of three emerging pollutants, pharmaceuticals, diclofenac, sulfamethoxazole, and cetirizine, on two samples of activated charcoal, WSCl2 and HWOH, was investigated. Textural analysis showed that HWOH has a prevalent mesoporous structure. On the other hand, WSCl2 was found to have a higher volume of micropores. In the case of all three pharmaceuticals, the higher removal efficiency was achieved using WSCl2. Different uptake of PhACs is linked with the different textural properties rather than the physicochemical properties of the pharmaceuticals. No direct correlation between removal efficiency and hydrophobicity/hydrophilicity or solubility was found. Adsorption data fitted the pseudo-second order kinetic model, which showed that adsorption was quicker on mesoporous HWOH, but WSCl2 had higher adsorption capacity. The uptake of pharmaceuticals from the mixture followed the same trends but overall achieved removal efficiencies were lower, visibly proving competitive adsorption between studied compounds.

**Author Contributions:** Conceptualization, L.Š., J.H., M.G. and M.N.; methodology, L.Š.; validation, D.P., L.F., P.H., E.M. (Emília Mališová) and E.M. (Eva Melníková); formal analysis, M.G.; investigation, D.P., L.F., P.H. and E.M. (Eva Melníková); data curation, D.P., P.H. and L.F.; writing—original draft preparation, D.P. and M.G.; writing—review and editing, M.G., M.N. and J.H.; visualization, D.P. and M.G.; supervision, L.Š. and J.H.; project administration, M.G. and M.N.; funding acquisition, J.H. and M.G. All authors have read and agreed to the published version of the manuscript.

**Funding:** This research was supported by Ministry of Education, Science, Research and Sport of the Slovak Republic under the project VEGA 1/0343/19. The research was also supported by Norway through the Norway Grants; Project: "Innovative carbon-based sorbents as an effective method of wastewater treatment," grant number 3213200008.

**Data Availability Statement:** The data presented and analyzed in this study are available on reasonable request from the corresponding author.

**Conflicts of Interest:** The authors declare no conflict of interest.

## Abbreviations

| | |
|---|---|
| AC | activated charcoal |
| AOP | advanced oxidation process |
| BET | Brunauer–Emmett–Teller |
| CET | cetirtizine |
| DAD | diode array detector |
| DCF | diclofenac |
| FTIR | Fourier transform infrared spectroscopy |
| HPLC | high-performance liquid chromatography |
| PhACs | Pharmaceutically active compounds |
| pKa | acid dissociation constant |
| SMX | sulfamethoxazole |
| SEM | scanning electron microscopy |
| SD | standard deviation |

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
