# Peer review of "Removal of Environmentally Harmful and Hardly Degradable Pharmaceuticals Sulfamethoxazole, Diclofenac, and Cetirizine by Adsorption on Activated Charcoal"

_water, doi:10.3390/w14243988_

Round 1

Reviewer 1 Report

Comments:

1- The authors should be provide an abbreviation list.

2- The title should be revise. In the title the sulfamethoxazole, diclofenac, and citrazine should be provide. Because the authors just have been worked with the these drugs.

3- The novelty of the proposed system would be presented in the introduction section. The adsorption of hardly degradable pharmaceuticals into activated charcoal is not a good novelty.

4- Advantage and disadvantages of the acivated charcol would be provide in the introduction section.

5- The adsorption isotherms such as Langmuir, Freundlich... should be provided in the manuscript.

6- How many times the proposed absorbents can be used in the batch system and what is their saturation values?

7- In order to better interpret the obtained results,the authors should be provide the results of SEM, EDX and FTIR of the adsorbents.

Author Response

Comments:

1- The authors should be provide an abbreviation list.

An abbreviation list has been added at the begining of the manuscirpt.

AC                  activated charcoal

AOP                advanced oxidation process

BET                Brunauer–Emmett–Teller

CET                cetirtizine

DAD               diode array detector

DCF                diclofenac

HPLC              high-performance liquid chromatography

PhACs             Pharmaceutically active compounds

pKa                 acid dissociation constant

SMX               sulfamethoxazole

2- The title should be revise. In the title the sulfamethoxazole, diclofenac, and citrazine should be provide. Because the authors just have been worked with the these drugs.

Thank you very much for your advice. We have modified the title of the manuscript.

Removal of Environmentally Harmful and Hardly Degradable Pharmaceuticals sulfamethoxazole, diclofenac, and cetirizine, by Adsorption on Activated Charcoal

3- The novelty of the proposed system would be presented in the introduction section. The adsorption of hardly degradable pharmaceuticals into activated charcoal is not a good novelty.

We agree with Reviewer that utilization of activated charcoal is not new process. However, there are some discrepancies in the literature among obtained results as stated in the Introduction (and discussed in the manuscript) and therefore elucidation of association of above-mentioned parameters is important for the future environmental applications. We have added this statement inti Introduction part of the manuscript.

4- Advantage and disadvantages of the acivated charcol would be provide in the introduction section.

Overall, there are many benefits to utilizing activated carbon to remove pharmaceuticals from waste streams. It is an easy, highly effective, and ecologically friendly approach that may be used in batch as well as continuous processes. However, higher manufacturing costs and a frequent requirement for regeneration could limit its overall effectiveness.

5- The adsorption isotherms such as Langmuir, Freundlich... should be provided in the manuscript.

Reviewer is right that these isotherms can be useful for the basic characterization of the system. However, in our work we focused on the practical application of two types of activated charcoal for the kinetic study of the adsorption of emerging pharmaceuticals from the wastewater. Moreover, as stated above, some different results compare to our observations are found in the literature concerning basic kinetics parameters of the system. Therefore, we have focused on this part of the problem and do not perform detailed basic thermodynamical research, although we agree that this could bring some uselfull information.

6- How many times the proposed absorbents can be used in the batch system and what is their saturation values?

Reusability of the carbons was not tested in this part of the research. The saturation values are displayed in the Table 5 as qe.

7- In order to better interpret the obtained results, the authors should be provide the results of SEM, EDX and FTIR of the adsorbents.

Thank you very much for this comment. Of course, we also did all the proposed experiments (instead of EDX we performed XRD analysis which is, in our opinion more useful for charcoal texture characterization), but since we obtained the most important information from the BET analysis, we did not include them in the original manuscript. Based on the reviewer's comment, we added a comment to the text regarding FTIR and XRD analysis of both activated charcoal samples and SEM images of both samples. All these analyzes confirm our results obtained from BET measurements.

Reviewer 2 Report

  1. Introduction part, if possible, some important and relative reports about Photocatalysis could helped: https://doi.org/10.1016/j.jtice.2021.08.034, https://doi.org/10.1007/s10904- 022-02389-8, doi: 10.5004/dwt.2019.24812, https://doi.org/10.1016/j.heliyon.2022.e09652.
  2. The novelty needs to refinement and should be highlighted in the introduction part.
  3. Authors need to spell English language throughout the manuscript. English is good, but some phrases are written with wrong word order.
  4. Why did the authors use the sulfamethoxazole, diclofenac?

55. I think that the authors need to make a deep characterization of their materiels used such FTR, EDX, XRD, BET etc.

66. the authors have to perform a kinetic study by the first and second order model to determine the limiting step of the kinetics

  1. It will be beter if the thermodynamic study has bee doing
  2. The authors should add the probal mechanism of adsorption of the three pollutants with the adsorbent material.

99.It is preferable that the thermodynamic study has been done.

   10. Table 2-6: no statistics and no standard deviations

11. The conclusion is also not targeted to the important aspects described in the manuscript; please rephrase it.

Author Response

Comments and Suggestions for Authors

  1. Introduction part, if possible, some important and relative reports about Photocatalysis could helped: https://doi.org/10.1016/j.jtice.2021.08.034, https://doi.org/10.1007/s10904- 022-02389-8, doi: 10.5004/dwt.2019.24812, https://doi.org/10.1016/j.heliyon.2022.e09652.

Thank you very much for your advice. We have added two of the suggested references to the manuscript.

  1. The novelty needs to refinement and should be highlighted in the introduction part.

We agree with Reviewer that utilization of activated charcoal is not new process. However, there are some discrepancies in the literature among obtained results as stated in the Introduction (and discussed in the manuscript) and therefore elucidation of association of above-mentioned parameters is important for the future environmental applications. We have added this statement inti Introduction part of the manuscript.

  1. Authors need to spell English language throughout the manuscript. English is good, but some phrases are written with wrong word order.

Thank you very much. We have corrected the grammar of the manuscript.

  1. Why did the authors use the sulfamethoxazole, diclofenac?

In our work, we have studied the adsorption of diclofenac, sulfamethoxazole, and cetirizine as these are among the most detected PhACs in the environment.

  1. I think that the authors need to make a deep characterization of their materiels used such FTR, EDX, XRD, BET etc.

Thank you very much for this comment. Of course, we also did all the proposed experiments (instead of EDX we performed XRD analysis which is, in our opinion more useful for charcoal texture characterization), but since we obtained the most important information from the BET analysis, we did not include them in the original manuscript. Based on the reviewer's comment, we added a comment to the text regarding FTIR and XRD analysis of both activated charcoal samples and SEM images of both samples. All these analyzes confirm our results obtained from BET measurements.

  1. the authors have to perform a kinetic study by the first and second order model to determine the limiting step of the kinetics

Kinetic study has been already performed in the original manuscript.

  1. It will be beter if the thermodynamic study has bee doing

Reviewer is right that thermodynamic studies can be useful for the basic characterization of the system. However, in our work we focused on the practical application of two types of activated charcoal for the kinetic study of the adsorption of emerging pharmaceuticals from the wastewater. Moreover, as stated above, some different results compare to our observations are found in the literature concerning basic kinetics parameters of the system. Therefore, we have focused on this part of the problem and do not perform detailed basic thermodynamical research, although we agree that this could bring some useful information.

  1. The authors should add the probal mechanism of adsorption of the three pollutants with the adsorbent material.

The pseudo-second order kinetics is typically connected with the cases where the rate of the direct adsorption process controls the total sorption kinetics. This information was added to the manuscript.

  1. Table 2-6: no statistics and no standard deviations

Thank you very much for this comment. We have added SDs to the respective Tables and information that all pH measurements were perfomed with SD=±2%

  1. The conclusion is also not targeted to the important aspects described in the manuscript; please rephrase it.

The conclusion was rephrased as follows:

“The adsorption of three emerging pollutants, pharmaceuticals diclofenac, sulfamethoxazole, and cetirizine, on two samples of activated charcoal, WSCl2 and HWOH, was investigated. Textural analysis showed that HWOH has a prevalent mesoporous structure. On the other hand, WSCl2 was found to have a higher volume of micropores. In the case of all three pharmaceuticals, the higher removal efficiency was achieved using WSCl2. Different uptake of PhACs is linked with the different textural properties rather than physicochemical properties of the pharmaceuticals. No direct correlation between removal efficiency and hydrophobicity/hydrophilicity or solubility was found. Adsorption data fitted the pseudo-second order kinetic model, which showed that adsorption was quicker on mesoporous HWOH, but WSCl2 had higher adsorption capacity. The uptake of pharmaceuticals from mixture followed the same trends but overall achieved removal efficiencies were lower, visibly proving competetive adsorption between studied compounds.”

Round 2

Reviewer 1 Report

The authors have addressed my concerns appropriately, and I recommend the manuscript be now accepted for publication in its current form.

Reviewer 2 Report

accepted in present form